# Research on the Mechanism of Collaborative Value Co-Creation of Enterprise–Science Community: A Case Study Based on the Green Brand Maoduoli

**Wenwen Shen** [1], **Yuankun Nie** [1], **Chao Long** [2], **Zibo Song** [3], **Qian Zhang** [4,*] and **Decai Tang** [5]

1   Business School, Yunnan University of Finance and Economics, Kunming 650221, China
2   Institute of Finance, Yunnan University of Finance and Economics, Kunming 650221, China
3   Yunnan Mao Duo Li Group Food Co., Ltd., Kunming 650221, China
4   Business School, Nanjing Xiaozhuang University, Nanjing 211171, China
5   School of Management Science and Engineering, Nanjing University of Information Science & Technology, Nanjing 210044, China
*   Correspondence: zhangqian@njxzc.edu.cn

**Abstract:** With the rapid increase of market competition pressure, enterprises' collaborative innovation plays a more prominent role in competitive advantage. This paper aims to explore how the enterprise–science community can achieve sustainable collaborative value co-creation. Taking the Maoduoli Group as a sample, using the single case study method and grounded theory, a structural model of the enterprise–science community collaborative value co-creation mechanism is constructed. The proposed model is based on the value logic of "advocating value—creating value—delivering value—acquiring value", which explains how the enterprise–scientific community collaborative value co-creation model is formed, how it is implemented, how it is delivered to customers, and the overall process of jointly harvesting value at last. The findings are as follows: First, government support, market demand, and entrepreneurial spirit are the internal and external factors for the enterprise–science community to develop collaborative value co-creation; secondly, the synergy mechanism of the enterprise–science community is to realize mutual activities such as joint research and development, a joint publication of papers, sharing of research results, joint research and development activities, and joint teaching practice through means of capital investment, concept support, and technical support. Third, the synergy mechanism of the enterprise–science community can realize the value of the economic and scientific research and the ecological and social benefits (narrow sense), and continuously feed back to further promote a deeper level of collaborative value co-creation of the enterprise–science community. This paper introduces the dimension of the scientific community, forms a special construct, and focuses on the collaborative value co-creation model of the enterprise–scientific community, which fills the gap in this research direction, and also provides theoretical support and practical guidance for the collaborative value co-creation model of the enterprise–scientific community.

**Keywords:** to create value; scientific community; synergy mechanism; a case study

## 1. Introduction

The sudden outbreak of COVID-19 has dealt a strong blow to the global economy. The enterprise model and development path that enterprises rely on are destroyed, and export setbacks, weak domestic demand, and natural disasters have led to a marked slowdown in economic growth. They desperately need to improve their competitive edge to meet the needs of the market; as a result, companies have to integrate more resources from outside and create knowledge through the collaboration of the enterprise–science community to improve the competitive advantage of enterprise innovation [1–3]. As a result, the cooperative value co-creation mode of the enterprise–science community has great research and reference significance.

The existing research on enterprise collaborative value co-creation mostly focuses on the competition and cooperation between companies with the intention of value creation [4], inter-firm innovation performance (namely collaborative innovation performance) [5–8], the influence mechanism of inter-firm cooperation and innovation performance [9–13], alliance innovation and collaborative innovation [14–16], knowledge collaborative innovation [17], and cooperation and innovation between enterprises and external innovation subjects [18]. Although the existing literature mentions the cooperative value co-creation of the enterprise–science community, there are few studies for this topic. Moreover, few of them focus on the cooperative value co-creation model of the enterprise–science community from the perspective of case studies and carry out in-depth exploration. Secondly, under the background of the sharp increase of market competition pressure, it is also an important gap for the enterprise–science community to achieve mutual benefit and a win-win situation through the optimized coordination mechanism.

As a result, based on the existing literature gap, the research question focuses on two aspects: first, how the enterprise–science community achieves mutual benefit and a win-win scenario through the synergy mechanism; secondly, how to introduce the dimension of the scientific community and construct the structural framework of enterprise–scientific community collaborative value co-creation. Through the selection of the green brand Maoduolii Group as the research object, using the single case study method based on grounded theory, we explore the enterprise–science community synergy value co-creation model in depth.

## 2. Related Literature Review

### 2.1. Research on Collaborative Innovation Theory of Enterprise–Science Community

Synergy, in English, means joint action, in early academic circles. Collaboration can help enterprises expand their fields according to their advantages, complete the matching of individual advantages and the overall environment, and finally make profits at the same time [19]. For example, companies can achieve synergy through shared skills, physical resources, coordinated strategies, vertical integration, negotiations with suppliers, and combined forces.

In recent years, related research contents of synergy theory include enterprise collaboration and expansion into several areas, such as enterprise and social sustainability [20], University–Industry collaboration [21], and enterprise and external collaborative research and development. University–Industry collaboration is a method of social cooperation or a voluntary effort made by industrial entities and educational and research institutions to solve problems or issues of common interest cooperatively [22]. University–Industry collaboration includes practical teaching, joint research and development activities, contract research, research and development consulting, innovation cooperation, a joint publication of papers, subject research, professional knowledge or research results sharing, and other interactions [21].

Collaborative innovation between enterprises and suppliers, customers, competitive enterprises, government, research organizations, intermediaries, etc., can effectively improve the technological innovation ability of enterprises. However, the enterprises with advantages in technological innovation are better than other enterprises in acquiring new technological knowledge [23–25]. There are two main drivers of enterprise technology research and development. One is to raise technical barriers and consolidate competitive advantages; the second is to spread the risk of research and development, speed up research and development, and form technical complementarity [26].

"Scientific community" was first proposed in 1942 by the British philosopher of science, M.Polanyi. It refers to the general abstract existence of a group of scientists who engage in scientific activities according to practical principles and standards under the constraints of common scientific norms and self-identification [27]. The internal system of the scientific community mainly includes the forms of scientific schools and invisible colleges. The social structure outside science comprises scientific societies, associations, research institutes,

and research institutions established by the government and society, which are the most extensive social organizations among the forms of the scientific community [27,28].

Taking the above into consideration, we find that universities, research organizations, research institutions, etc., all belong to the scientific community. In the process of collaborative innovation of the enterprise/industry–science community, continuous scientific research capacity is created, peer review and academic exchange of scientific results are promoted, and social value for applying scientific research results is created [29].

### 2.2. Research on Value Co-Creation of Enterprise–Science Community

The theory of value co-creation was developed by Prahalad and Ramaswamy, and most current research is based on two logics: consumer-experience-dominated logic [30] and service-dominated logic [31]. The former focuses on the realization of value for all parties [32]; it is the value co-creation mode of direct interaction between enterprises and consumers. The latter emphasizes the value realization of the value co-creation system [32]; it is a value co-creation mode that integrates and utilizes the resources of all participants (organizational ability, personal skills, knowledge, and quality and quantity of key stakeholders, etc.) [33,34].

All stakeholders may be regarded as resource integrators of value co-creation [35]. Value co-creation bodies gradually expanded to multi-stakeholder groups, enterprise ecosystems, etc. [36–38].

Stakeholders are those who can influence the realization of corporate goals or any individual or group that can be affected by the process of achieving the goals of the enterprise [39]. Therefore, the scientific community is also a part of the stakeholder group. The sharing of key resources (human resources, knowledge, skills, experience, materials, funds, etc.) between the scientific community and enterprises is conducive to the cooperative development of the scientific community and enterprises. Funds are the economic benefits sought by the scientific community, while knowledge and technology are the support for enterprises to enhance their core competitiveness. The interaction and integration of resources between the two are the common needs and expectations of both sides [21].

Combining the above studies, the enterprise–science community is based on the value logic of "advocating value—creating value—delivering value—acquiring value" [40,41], driven by extrinsic motivation and underlying economic factors, such as reputation improvement, experimental research [42], sharing of key resources, enhancing competitive advantage, and capital benefits [21], through exchange and resource integration to achieve value co-creation [43,44].

### 2.3. Literature Comments

From the above studies, it can be found that as technology plays a bigger role in enterprise operations, collaboration becomes the main way enterprises generate new technologies [45]. The focus of synergetic theory has changed over time. First, on the collaborative object, from the early enterprise, internal collaboration expanded to external collaboration and other fields [4–8,18]; second, in collaborative content, instead of being stuck on one product, it extends from the early by-products to all collaborative products (innovative technology) [17]; finally, in terms of synergistic objectives, we no longer seek short-term benefits, but focus on improving core competitiveness to achieve sustainable management [21].

However, under the background of deep integration of industry, university, and research institutes in China, several aspects of enterprise collaboration still need to be further discussed. 1. How to explore the synergistic mechanism between enterprise and science community. 2. In the context of industry–university–research cooperation, the scope of external innovation subjects has gradually expanded. How should enterprises seek the path of mutual benefit and win-win scenarios with the scientific community? How to effectively achieve the optimal coordination goal has thus become an urgent issue to discuss. 3. Few existing studies have deeply discussed the collaborative value co-

creation model between enterprises and scientific communities from a holistic perspective (cause–process–result). 4. There are few case studies for this purpose.

Therefore, this study follows the value logic of "advocate-value-creation—value-delivery—value-capture" [40,41], introduces the dimension of the scientific community, forms a special construct, and analyzes the collaborative value co-creation mode of the enterprise–scientific community to construct a more complete collaborative value co-creation structure framework and provide some reference for similar enterprises.

## 3. Research Methods

The purpose of this study is to discover how an enterprise–science community can achieve sustainable synergistic value co-creation. The problem it explores requires a comprehensive and systematic analysis of the enterprise collaborative mechanism and value co-creation. Given the extreme nature of the subjects and the complexity of the questions, this paper adopts the single-case multi-level research method, and provides an in-depth analysis of the phenomenon and situation based on grounded theory. It further digs into the deeper theoretical rules to make connections between constructs, thus, forming the corresponding theoretical framework and constructing the model of cooperative value co-creation of the enterprise and scientific communities.

### 3.1. Case Selection

Since Maoduoli was founded, focusing on the comprehensive research, development, and application of Yunnan's wild biological resources, it has adhered to the "four do not add" quality code, namely: without pigment, without flavor, no preservatives, without synthetic fragrances. It is a well-known green food brand in China. This article follows the extreme, revelatory, and typical principles, selecting the green brand "Maoduoli Group " as the case study object (herein after referred to as "Maoduoli"). The specific reasons are as follows: 1. Compared with the listed companies in the same food industry in China, their investment in scientific research accounts for a very high proportion, as shown in Table 1 (Data source: the 2016–2020 annual reports of each brand and the internal data of Maoduoli); 2. Maoduoli is the only company outside of Japan that has fully mastered the tamarind polysaccharide refining technology (data source: article on Maoduoli's official account) and the first company in China that has fully mastered the tamarind polysaccharide refining technology (data source: news report on "Yunnan.com"); 3. They have received many honors, such as: "Consumers' Favorite Food Brand (2005, 2009)", "China Famous Trademark (2014)", "Top 20 Innovative Green Food Enterprises in Yunnan Province (2019, 2020)", and "The Most Beautiful Green Food Enterprise (2020)"(data source: internal data provided by Maoduoli); they have high influence and high consumer love; 4. As an excellent teaching case, the practice of collaborative value creation between Maoduoli and the science community has been included in the well-known academic platform, China Management Case Sharing Center (Case No.: MKT-1061), which can be used as a teaching reference for enterprises and colleges and universities across the country. It is a model for Chinese enterprises.

Taking into account the above, the cooperative value co-creation model of Maoduoli and the science community (Yunnan Institute of Thermal Ecological Agriculture, Yunnan Institute of Horticultural Crops, the University of Shanghai for Science and Technology, etc.) is very typical and extreme; there are better models in China. As a local brand with great popularity and loyalty in Yunnan, it is more convenient to collect data, which is abundant, and it plays a key role in the in-depth case study.

**Table 1.** Well-known food brand research and development contrast.

| Region | Brand | Main Products | R&D Content (Data Source: 2020 Annual Report of Each Brand) | Proportion of Scientific Research Investment (Proportion of Scientific Research Investment = Research and Development Costs/Gross Operating Income * 100% (2016–2020)) | R&D Cycle |
|--------|-------|---------------|------------------------------------------------------------|---|-----------|
| China | Three squirrels | Snacks | Control of nut oil oxidation, preserving meat products with soy sauce, development and application of fermentation strains for leisure food | 0.42% | Long term, annual R&D |
| China | Langyuan Shares | Snacks, Baking materials | Expand the enterprise of smart city visualization platforms, data center life-cycle management R&D, smart city data ecological environment | 0.78% | Long term, annual R&D |
| China | Miss you very much | Jujube food | Food maintenance, food tonic, food therapy | 0.58% | Long term, annual R&D |
| China | Good shop | Snacks | Health nutrition research, research on processing technology | 0.41% | Long term, annual R&D |
| China | Maoduoli | Snacks | Tamarind fruit cake with high dietary fiber, Yunnan olive tablets, passiflora buccal tablet, tamarind polysaccharide, tamarind fruit cake with flowers | 4.49% | Long term, annual R&D |

Source: Author sorting.

*3.2. Data Source*

The data sources of this paper are a combination of primary and secondary data. Primary information includes: (1) Unstructured interviews from December 2020 to November 2021. Multiple in-depth interviews were conducted with the senior management of the Maoduoli Group; (2) Field investigation of scientific research bases in November 2021. Field research was conducted on Maoduoli Manor, including in-depth interviews with base managers and R&D staff; (3) Observation from December 2020 to January 2021 of some local supermarkets (Walmart, snack shop, etc.), and conducted on-site interviews with customers who bought Maoduoli products, which information was documented. Secondary information includes: (1) Internal data, such as strategic discussion data, enterprise annual report data, technology development contract, industry–university–research cooperation data, etc.; (2) Open external materials, such as Gigafactory video, news reports, journal papers, Baidu Baike, Weibo post bar, and other relevant materials. Specific descriptive statistics are shown in Table 2.

**Table 2.** Case Data Sources.

| Data Source | Interviewees | Interviews | Interview Duration/Min |
|-------------|--------------|------------|------------------------|
| Unstructured interview | Chairman | 4 times | five hundred and sixty-nine |
| | Chief Financial Officer | 3 times | four hundred and forty-nine |
| | Special assistance from the Director | 2 times | one hundred |
| | Bank President | 3 times | four hundred and forty-nine |
| | Other relevant employees | 3 persons | ninety-one |
| Base survey | Base Infrastructure Manager | 1 time | one hundred and thirty-one |
| | Base Scientific Research Manager | 1 time | one hundred and thirty-one |
| | Other relevant employees | 6 persons | one hundred and thirty-one |
| Participate in observation | Supermarket customers | 12 persons | |
| Internal data | Strategic discussion materials, annual report data of some enterprises, technology development contracts, materials of the tamarind development company, and materials of industry–university research cooperation (16 in total). | | |
| Public information | AV, news report, Baidu Encyclopedia, Weibo, Douyin, Zhihu, Tieba, and other platforms (40 copies in total). | | |

Source: Author sorting.

From December 2020 to January 2022, 15 senior management members of the case enterprise were interviewed in depth many times, and the interview outline is shown in Table 3.

**Table 3.** Interview outline.

| Interview Topic | Outline of Main Contents |
|---|---|
| Essential information | Gender, age, education background, family structure, company position |
| Cooperative value co-creation mechanism between enterprise and scientific community | ● Enterprise development history; <br> ● Enterprise management concept; <br> ● Enterprise development direction; <br> ● Enterprise industry–university research cooperation; <br> ● Enterprise scientific research capability. |

### 3.3. Data Analysis

Grounded theory (GT) is a kind of qualitative research approach proposed by American sociologists Anselm Strauss and Barney Glaser in their book "The Discovery of Grounded Theory" in 1967, defined as the discovery of theory from data. Compared to other research methods, grounded theory focuses more on combining theory and practice. The idea is that an in-depth analysis of the source material, based on empirical facts, gradually forms a theoretical framework. According to the theoretical framework of grounded theory, it can make research theory have practical use and significance and, finally, realize the knowledge and data exchange between academia and practice.

As a result, this paper adopts an open research perspective based on grounded theory, according to the value logic of "advocating value—creating value—delivering value—acquiring value". The unstructured interview and base survey were used for primary data [40,41], as shown in Figure 1 (the process of creating and delivering value is a process of interaction and collaborative creation between enterprises and scientific community resources), in turn from the open coding, spindle coding, and selective coding of three aspects of sorting and analysis.

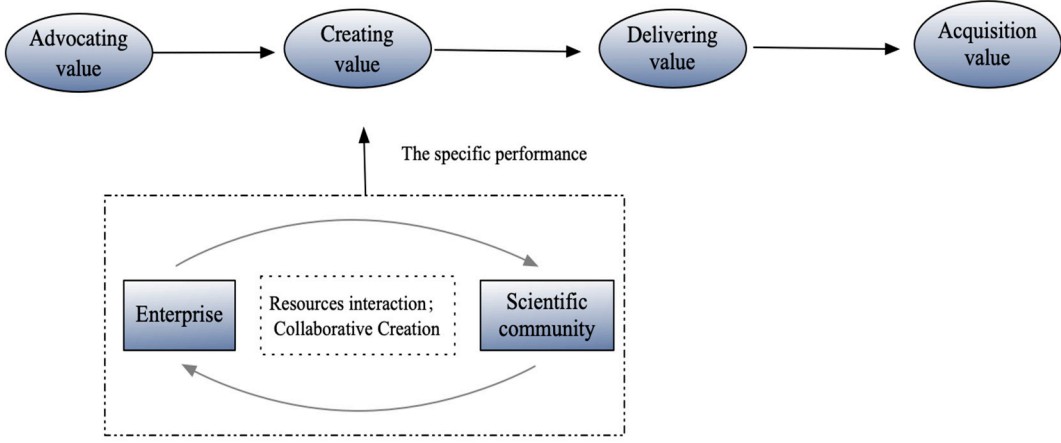

**Figure 1.** Case analysis logic.

### 3.3.1. Open Coding

The goal of open coding is to help researchers anchor phenomena, define concepts, and explore categories; that is, the concept is clustered. In this paper, native code is basically used in open coding, which is the original words the subjects used to express their opinions; this is conducive to a true reflection of the respondents' views. Table 4 embodies the process of conceptualizing and categorizing the original interview records in this paper. The result

of categorization is the form factor and value connotation of cooperative value creation between enterprises and the scientific community.

**Table 4.** The enterprise and the scientific community co-create the open coding process and results.

| Original Interview Sentence (Representative Statement) | Category |
| --- | --- |
| In 2020, we opened a Tamarind museum called Maoduoli, we will work with experts to explore the knowledge of tamarind intuitive display to consumers, let the vast number of consumers can close contact with our tamarind products, In-depth understanding of our product production process, deeply promote our corporate culture, Let everyone on our products to buy rest assured, eat rest assured. | Build a Tamarind museum; Impart knowledge of tamarind; Improve customer perceived value |
| Three thousand mu planting base, just to control it from the source, to make quality products, the production plant of Maoduoli is built according to GMP standards, the air in the inner workshop reaches the purification standard of 10 grades, this is also the key to ensure the quality of Maoduoli products under the condition of "four do not add". Maoduoli strictly follows Standard Operating Procedures for Food hygiene (SSOP), the qualified rate of products reaches 100%, the green standards implemented by enterprises are far beyond national standards. Strong research and development capabilities, advanced production equipment and environment, these are our confidence, the green standards implemented by enterprises are far beyond national standards. Strong research and development capabilities, advanced production equipment and environment, these are our confidence, based on this, we invested a lot of money, only to develop the pure ecological green product "tamarind fruit cake" without any pigment, flavor, preservative or artificial synthetic fragrance, make sure people eat healthy, the majority of consumers have been recognized and love. | Consumer health value |
| In 2017, we cooperated with the expert team of Allian Zhong, research and development of passion flower, various flavors of tamarind fruit cake products, fully satisfy the taste buds of consumers. | Improve consumer satisfaction |
| We beat drums and gongs in the supermarket to promote the green growth road of Maoduoli in the most simple way. | Special shopping and supermarket promotion Guiding green life consensus |
| Maoduoli has purchased extremely advanced green production equipment, and working with a number of experts, the research and development of tamarind extracting, in order to produce pure natural green products, get the recognition and love of consumers. | Improve corporate profitability |
| The 1999 World Expo brought us an opportunity, we developed Yunnan 18 strange products, Maoduoli has become the first company to combine Yunnan's regional culture with food, achieved fame and fortune. | Improve enterprise performance; Improve corporate social image |
| We started with the development of tamarind and then went on to passion fruit, Dian olive and Doi, based on our cooperation with the expert team of Allian Zhong, I hope to become a master of Yunnan fruit cooking, to attract consumers with a wide variety of fruit flavors. | Increase product attractiveness |
| In 2012, Maoduoli works with the Yunnan Academy of Agricultural Sciences, the high yield cultivation of tamarind fruit cake was achieved successfully, the cost of raw material of tamarind was reduced. | Reduce production costs |
| We have cooperated with the Institute of Thermal Eco-Agriculture of Yunnan Academy of Agricultural Sciences and Horticultural Crops Research of Yunnan Academy of Agricultural Sciences for many times, raw material protection, soil erosion control, barren hills transformation, we did all that, we want to achieve more than the survival of the enterprise, I hope I can contribute to the society and ecological restoration, make the existence value of Maoduoli more meaningful. | Increase brand value |

**Table 4.** *Cont.*

| Original Interview Sentence (Representative Statement) | Category |
| --- | --- |
| In 2012, we set up grassroots scientific research stations, and cooperated with Duan Yuetang, Sha Yucang and other top scientific experts, the cultivation techniques of dwarf dense planting, high-yield cultivation, flower and fruit stimulation, disease control and insect control were studied, and free of charge to provide dense dwarf, high yield cultivation and other planting technology, guide farmers to carry out scientific planting, in this way, more than 3000 farmers in the Honghe River Basin and Jinsha River basin planted more than 20,000 mu of tamarind and passion flower. | Increase raw material yield; Science and technology and industry help agriculture |
| In collaboration with the Yunnan Academy of Agricultural Sciences, the transformation of barren mountains has been successfully realized, we provided employment for more than 3000 local rural households to relieve government pressure and lift themselves out of poverty, promoted Xinping County agricultural economic development. | Increase local employment rate; Poverty alleviation through industry and science and technology; Promote local economic development |
| In 2004, Maoduoli Group and Yunnan Academy of Agricultural Sciences join hands, it's the beginning of a journey to preserve the ancient tamarind tree. Today, the average age of the trees protected by the Maoduoli Group is more than 200 years, nearly 60,000 plants, distributed in the Jinsha River basin and the Red River basin. The tamarind forest improves the harsh environment of the hot and dry valley, the intensity of surface runoff and soil erosion is alleviated obviously, many natural disasters caused by the loss of vegetation were significantly reduced in the tamarind forest. | Protect the resources of tamarind; Alleviating soil erosion; Reduce the rate of natural disasters |
| Tamarind belong to perennial species, five years bloom, seven years bear fruit, ten years yield. In order to increase the utilization rate of land and safeguard the benefits of fruit farmers, we cooperate with the Horticultural Crops Research Institute of Yunnan Academy of Agricultural Sciences, research on efficient understory compound planting technology was carried out, such as a variety of aquaculture center, fish pond, water and fertilizer integration of automatic irrigation, in order to achieve the same benefits in the growth cycle of tamarind. | Increase land utilization; Guarantee the income of fruit farmers |
| We work with the Hot Zone Eco-Agriculture Research Institute, a meteorological observation station and soil erosion observation were built in Maoduoli Manor, the collected resources should be reasonably, standardized and regionalized in the planting resource nursery, It provides scientific meteorological data and soil erosion data for improving local microclimate environment. | Improve microclimate environment |
| Based on the typical climate of its hot and dry valleys, select high-quality varieties of tamarind tree to carry out vegetation restoration and establish irrigation facilities, etc, strengthening investment in science and technology, deepen cooperation with Research Institute of Hot Area Eco-Agriculture, a standardized soil and water loss observation station has been established. Based on sloping terrain, using carbon sequestration technology, control soil erosion, the real data provided the basis for the study of soil carbon cycle in plantation, it is proved that the ecological benefit produced by the cultivation of artificial tamarind is far better than that of other economic forests.It has created immeasurable value for the ecological environment restoration of the Red River basin. | Control soil erosion; Restoration of the ecological environment |
| The tamarind germplasm nursery of Maoduoli has the most abundant resources in the country, and got the Guinness Book of Records, it is the result of the joint efforts of many tamarind cultivation experts, related leaders of Xinping County Party Committee and experts of provincial and municipal Academy of Agricultural Sciences. The purpose of this research is to explore the geographical distribution characteristics, shape variation trend, and species exploitation and utilization of the global tamarind resources, it has laid the material foundation for the development of the tamarind industry of Chinese sour horn, effectively promote the species protection and efficient utilization of tamarind, build the "chip" of modern agriculture. | Promote the development of tamarind industry; Promote scientific research and development in the field of tamarind; Protect agricultural germplasm resources; |

**Table 4.** *Cont.*

| Original Interview Sentence (Representative Statement) | Category |
|---|---|
| Based on the transformation of barren mountains, we set up Maoduoli Manor with the Research Institute of Yunnan Academy of Agricultural Sciences as the core, signed expert workstations. The research focuses on ecological restoration (soil and water loss control), planting and cultivation of tamarind, understory complex ecology, and planting resource nursery. Maoduoli has reached a strategic cooperation with the top food research expert team in China—Professor Ai Lianzhong's team from University of Shanghai for Science and Technology, the expert workstation of Ailian has been established, The research and development of the seeds of tamarind were emphasized, to realize the transformation of tamarind polysaccharide and other scientific research achievements. | Promote the transformation of scientific research achievements; Ensuring continuous scientific research capability; Promote the coordinated development of scientific research teams, research institutions and enterprises |

Source: Author sorting.

### 3.3.2. Spindle Code

Principal axis coding can show the organic relationship between the various parts of the data; the principal axis coding follows the principle that a theory is sufficiently significantly correlated with the core variable [46], finding clues to the construction of the theory, then a structural analysis framework is formed [47].

Based on the interview materials and the original interview sentences in Table 4, we find that the enterprises first cooperated with the scientific community under the pressure of raw materials and with the help of the government. When the results of cooperation are significant, enterprises begin to voluntarily cooperate deeply with the scientific community, which is the evolution of "value proposition". After that, through investment in the scientific community, joint research and development of the subject, joint publication of papers, professional knowledge sharing, and other interactions, the innovative competitive advantage of the enterprise is further expanded, which is the process of creating value. Then, with the tamarind museum and its ultra-propaganda characteristic, it implements straightforward value delivery and gradually gains value (social, ecological, economic, and scientific research) in the end. And the value acquired continues to feed back into the business and scientific community, further deepening the synergy between the business and scientific community and realizing more value. As shown in Tables 5 and 6.

**Table 5.** Spindle coding process and results.

| Connotation of Categorical Relations | Corresponding Subcategory | Main Category |
|---|---|---|
| The cooperative value co-creation of the enterprise and scientific community is the result of entrepreneurship and government support. It is the proposition value in the logic of collaborative value co-creation. | Entrepreneurship; Government support | Advocating value |
| The cooperative value co-creation behavior of the enterprise and scientific community is the result of catering to the market demand. It is the process of creating value in the logic of collaborative value co-creation. | Market demand | Creating value |
| The cooperative value co-creation behavior of the enterprise and scientific community is the result of the large amount of enterprise funds supporting the scientific community. It is the process of creating value in the logic of collaborative value co-creation. | Capital investment | |

**Table 5.** *Cont.*

| Connotation of Categorical Relations | Corresponding Subcategory | Main Category |
|---|---|---|
| The cooperative value co-creation behavior of the enterprise and scientific community has a result that the scientific community provides technical support for enterprise. It is the process of creating value in the logic of collaborative value co-creation. | Technical support | |
| The cooperative value co-creation behavior of the enterprise and scientific communities is the result of their continuous expansion of scientific research fields, in-depth exploration of scientific research achievements, and continuous resource sharing. It is the process of creating value in the logic of collaborative value co-creation. | Resource sharing (research and development, joint publication of papers, sharing of research results, joint research and development activities, joint teaching practice, etc.) | |
| The collaborative value of the enterprise and scientific community co-creates and feeds the collaborative development of both. | Promote the coordinated development of scientific research teams, research institutions and enterprises | |
| The construction of Cape Sour Museum is a direct way to convey customer value and knowledge value. | Build a Tamarind museum | |
| Imparting the knowledge of tamarind is a direct way to transfer the value of knowledge. | Impart knowledge of tamarind | Delivering value |
| The promotion of special stores and supermarkets is a direct way to transmit brand culture and an intuitive presentation of value. | Special shopping and supermarket promotion | |
| The science and technology industry helps agriculture in the social benefit (narrow sense) dimension in the cooperative value creation. | Science, technology, and industry help agriculture | |
| Increasing the employment rate is the social benefit (narrow sense) dimension in the co-creation of collaborative value. | Increase local employment rate | |
| Promoting the development of local economy is the dimension of social benefit (narrow sense) in the co-creation of collaborative value. | Promote local economic development | Acquisition value—Social benefits (narrow sense) |
| Poverty alleviation in the science and technology industry is the social benefit (narrow sense) dimension in the co-creation of collaborative value. | Poverty alleviation through industry and technology | |
| Guiding the green life consensus is the social benefit (narrow sense) dimension in the collaborative value co-creation. | Guiding green life consensus | |
| Guaranteeing fruit farmers' income is the social benefit (narrow sense) dimension in the cooperative value creation. | Guarantee the income of fruit growers | |
| Improving enterprise profitability is the economic benefit dimension of the co-creation of collaborative value. | Improve enterprise profitability | |
| Improving enterprise performance is the dimension of economic benefit in the co-creation of collaborative value. | Improve enterprise performance | Acquisition value—Economic performance |
| Improving the social image of enterprises is the economic benefit dimension in the co-creation of collaborative value. | Improve corporate social image | |
| Customer-perceived value is the economic benefit dimension in the co-creation of collaborative value. | Improve customer-perceived value | |

**Table 5.** *Cont.*

| Connotation of Categorical Relations | Corresponding Subcategory | Main Category |
|---|---|---|
| Consumer health value is the economic benefit dimension in the co-creation of collaborative value. | Consumer health value | |
| Improving consumer satisfaction is the economic benefit dimension of collaborative value co-creation. | Improve consumer satisfaction | |
| Increasing product attraction is the economic benefit dimension in the co-creation of synergistic value. | Increase product attractiveness | |
| Reducing production cost is the economic benefit dimension in the co-creation of collaborative value. | Lower production cost | |
| Increasing brand value is the economic benefit dimension of the co-creation of collaborative value. | Increase brand value | |
| Improving raw material output is the economic benefit dimension in the co-creation of synergistic value. | Increase the output of raw materials | |
| Promoting the development of the tamarind industry is the economic benefit dimension in the co-creation of synergistic value. | Promote the development of the tamarind industry | |
| Conservation of tamarind resources is the dimension of ecological benefit in the co-creation of synergistic value. | Protection of tamarind resources | |
| Promoting the conservation of agricultural germplasm resources is the dimension of ecological benefit in the co-creation of cooperative value. | Protection of agricultural germplasm resources | |
| Alleviating soil and water loss is the dimension of ecological benefit in the co-creation of collaborative value. | Mitigate water and soil loss | |
| Reducing the rate of natural disasters is the dimension of ecological benefits in collaborative value creation. | Reduce natural disaster rate | Acquisition value—Ecological benefit |
| Increasing land utilization rate is the dimension of ecological benefit in the co-creation of collaborative value. | Increase land utilization rate | |
| Improving microclimate environment is the dimension of ecological benefit in the co-creation of collaborative value. | Improve microclimate environment | |
| Soil and water loss control is the dimension of ecological benefit in the co-creation of cooperative value. | Control water and soil loss | |
| Restoration of the ecological environment is the dimension of ecological benefit in the co-creation of collaborative value. | Restoration of ecological environment | |
| Promoting the development of scientific research in the field of tamarind is the dimension of scientific benefit in the co-creation of collaborative value. | Promote the development of the scientific research field of tamarind | |
| Ensuring sustainable scientific research capacity is the dimension of scientific research benefit in the co-creation of collaborative value. | Ensuring continuous scientific research capability | Acquisition value—Scientific research benefits |
| Promoting the transformation of scientific research achievements is the dimension of scientific research benefit in the co-creation of collaborative value. | Promote the transformation of scientific research achievements | |

Source: Author sorting.

**Table 6.** Selective coding result.

| Connotation of Relationship Structure | Typical Relationship Structure | Core Category |
|---|---|---|
| Entrepreneurship is the internal driving factor of collaborative value co-creation between enterprises and the academic community, it is the premise of collaboration between enterprise and scientific community | Entrepreneurship → creation of value → transfer of value → acquisition of value | |
| Government support is an external driving factor for enterprises and the academic community to create collaborative value, and is the premise for enterprises and the scientific community to collaborate | Government support → entrepreneurship → creation of value → transfer of value → acquisition of value | |
| The market demand is the external restriction factor of the cooperative value co-creation between enterprise and scientific community; it is the premise of collaboration between enterprises and the scientific community | Market demand → Enterprise and scientific community synergy | |
| Government support is the initial driving factor for the collaborative value creation of enterprises and scientific communities, and is the premise of the collaboration between enterprises and scientific communities | Government support → entrepreneurship → creation of value → transfer of value → acquisition of value | |
| The enterprise's capital investment and concept support to the scientific community, the scientific community's technical support to the enterprise, and the resource sharing between the two are the collaborative path of the enterprise and the scientific community, the necessary process to realize the value co-creation, and can directly affect the result of the value co-creation | Enterprise → capital input → scientific community → transfer value → obtain value Enterprise → idea support → scientific community → transfer value → obtain value Scientific community → technical support → enterprise → transfer value → obtain value Enterprise → Resource sharing research and development, joint publication of papers, research results sharing, joint research and development activities, joint teaching practice, etc. ← Scientific Community → transfer value → Obtain value | Cooperative value co-creation mechanism (Value proposition → value creation → value delivery → value acquisition) |
| The construction of a tamarind museum, knowledge sharing and characteristic publicity are the direct expression of the cooperative value creation between enterprises and scientific communities, and the direct way of transferring value | Delivering value (building Sour Kok Museum, knowledge sharing, corporate promotion) | |
| Social benefit (narrow sense), economic benefit, ecological benefit, and scientific research benefit are the final value content obtained by the collaborative value co-creation of enterprises and scientific communities, and are the foundation for the continuous collaborative value co-creation of enterprises and scientific communities | Obtain value (social benefit (narrow sense), economic benefit, ecological benefit, scientific research benefit) → enterprise and scientific community synergistic value co-creation | |

Note: The arrows (→) in this table are explained initially in the table "Connotations of Relationship Structure" and in detail in Section 4 "Case Studies and Findings".

### 3.3.3. Selective Coding

Selective coding is more selective, directional, and conceptual [46]; its category has more subtle and complete characteristics. After open coding and spindle coding of the primary data in the interview part, comparing, elaborating, and interpreting the theoretical framework, selective coding is carried out again to form the theoretical rudiment. To sum

up, government support is the enterprise with scientific community collaborative value creating external driving factors. The market demand is the enterprise with scientific community collaborative value creating external constraints. The behavior of an enterprise with voluntary in-depth cooperation shows that entrepreneurship is to deepen the enterprises and the scientific community's internal driving factors. Value proposition → value creation → value delivery → value acquisition is the mechanism of value co-creation between enterprises and the scientific community, as shown in Table 6.

### 3.3.4. Saturation Test

According to the criterion of "theoretical saturation", in this study, interview data and raw data collected from multiple platforms (Douyin, Weibo, Baidu, etc.) (about 1/3 of the total samples) were used for the theoretical saturation test. As it turns out, the categories in the model are rich enough; no new important categories and relationships were found. It can be concluded that the above structure of cooperative value co-creation between enterprises and scientific communities is theoretically saturated.

## 4. Case Analysis and Discovery

The coding framework is based on primary and secondary data. On the one hand, through bilateral data from businesses and consumers as primary and secondary multilateral data, the rationality of the co-creation mechanism model of enterprise and collaborative value is verified. On the other hand, it also proves that selecting the green brand "Maoduoli Group" as a case sample is extremely appropriate and accurate. The following is an in-depth study and analysis of the case of the Maoduoli Group and a statement of valuable findings.

### 4.1. Case Summary

Maoduoli is involved in raw material cultivation, product development, production, and sales as one of the modern food-intensive processing enterprises. Maoduoli has worked with the scientific community since its inception. Several scientific research platforms have been created, such as the Yuxi expert basic scientific research workstation, Yunnan expert basic scientific research workstation, Yunnan Academician (expert) workstation, postdoctoral research workstation, etc. (source: Maoduoli internal data). So far, Maoduoli has created 126 patents in collaboration with the University of Shanghai for Science and Technology, Yunnan Academy of Agricultural Sciences Hot Area Agricultural Graduate Institute, and Shanghai Nightgrass Biotechnology Co., Ltd., of which 20 are invention patents, 14 are utility model patents, and 100 are design patents (source: internal information provided by Maoduoli).

Maoduoli and the scientific community have created a number of utility values since 2012, when they completed the "Barren Mountain Transformation Research Base" in collaboration with the Yunnan Institute of Thermal Eco-Agriculture and Yunnan Institute of Horticultural Crops. 1. The land utilization rate of over 3,000 mu in Mosha Town, Xinping County, Yuxi City, Yunnan Province has increased from 0% to 100% (data source: Maoduoli internal information and Photos taken during in-depth interviews with research sites). 2. They achieved high-yield cultivation cooperation, that is, the fruit hanging date of tartar was advanced by 3 years, and the raw material cost of tartar was reduced by 8.5% compared with the previous month (source: Maoduoli internal data (2012)). 3. They promoted the development and utilization of keratin from seed, seed coat, seed kernel, and fiber, and the application of keratin, antioxidants, and polysaccharides in food, drugs, cosmetics, and other fields (source: Maoduoli internal information). 4. They provide planting technology such as dense planting dwarfization and high-yield cultivation for free, guide farmers to carry out scientific planting, and drive more than 3000 farmers in Honghe and Jinsha River basins to plant more than 20,000 mu of Acanthus and passionflower. They have solved the employment problems of more than 3000 local farmers and helped Mosha Town, Pingdian Town, Pingzhang Town, and other areas in Xinping County get rid of poverty and become

rich. Thus, they achieved poverty alleviation through science and technology and industry (data source: "Sohu.com" news report). 5. They established standardized soil and water loss observation stations to achieve soil and water loss control and ecological restoration by increasing carbon sinks and other means (data source: Photos taken during in-depth interviews with research sites). 6. After the collaboration with the University of Shanghai for Science and Technology, the revenue increased by 10% qO-m and the profit increased by 15% QO-m. As of 2021, Maoduoli's products have reached 80% market share in Yunnan Province (source: Maoduoli internal information and "Sohu.com" news report).

### 4.2. Enterprise–Science Community Synergy Mechanism Formation Path

Taking the case of Maoduoli into account, we find that Dori was initially forced by external market demand and raw material shortage and, with the government's support, collaborated with the scientific community. When working with the scientific community to achieve significant economic benefits (8.5% reduction in raw material costs, 100% improvement in land utilization, 10% QM increase in revenue, 15% QM increase in profits, etc.), Maoduoli leads with an entrepreneurial spirit, with an annual research investment ratio of up to around 4% (source: Internal information provided by Maoduoli), provides a large amount of financial support to the scientific community (far more than the proportion of scientific research investment of listed companies in the Chinese food industry, as shown in Table 1), and deepens the cooperation with the scientific community, which is the evolution of the "proposition value" in the collaborative value creation process between Maoduoli and the scientific community. Among them, government support is the external driving factor for the collaborative value co-creation between enterprises and the scientific community, market demand is the external restricting factor for the collaborative value co-creation between enterprises and the scientific community, and the voluntary and in-depth cooperation behavior of enterprises thereafter indicates that entrepreneurship is the internal driving factor for deepening the collaborative value co-creation between enterprises and the scientific community.

Since then, Maoduoli and the scientific community have carried out joint research and development, joint publications, research result sharing, joint research and development activities, joint teaching practices, and other interactive activities (source: Maoduoli internal information). This is the process of "creating value" to further expand the innovative competitive advantage of enterprises. From the formation factor (value proposition) to the collaborative process (value creation), we sorted out the formation path of the collaborative mechanism between enterprises and the scientific community, as shown in Figure 2.

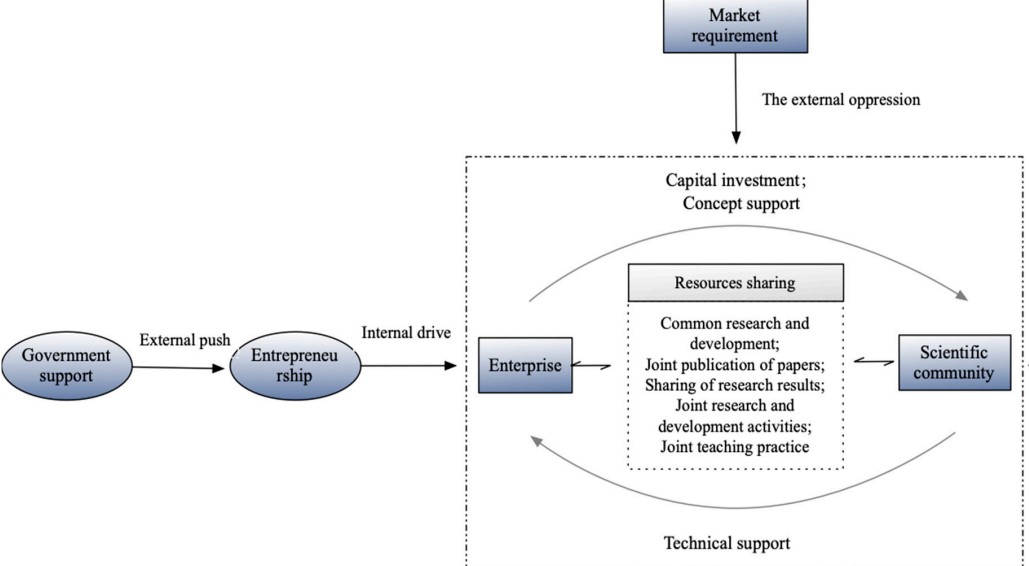

**Figure 2.** Enterprise–science community synergy mechanism formation path.

### 4.3. Enterprise–Scientific Community Synergy Value Co-Creation Mechanism

Based on Figure 2 and the value logic of "value proposition → value creation → value delivery → value acquisition", we constructed the structural framework of the collaborative value co-creation mechanism of the enterprise–scientific community, as shown in Figure 3, and analyzed and refined the collaborative value co-creation mechanism of the enterprise–scientific community.

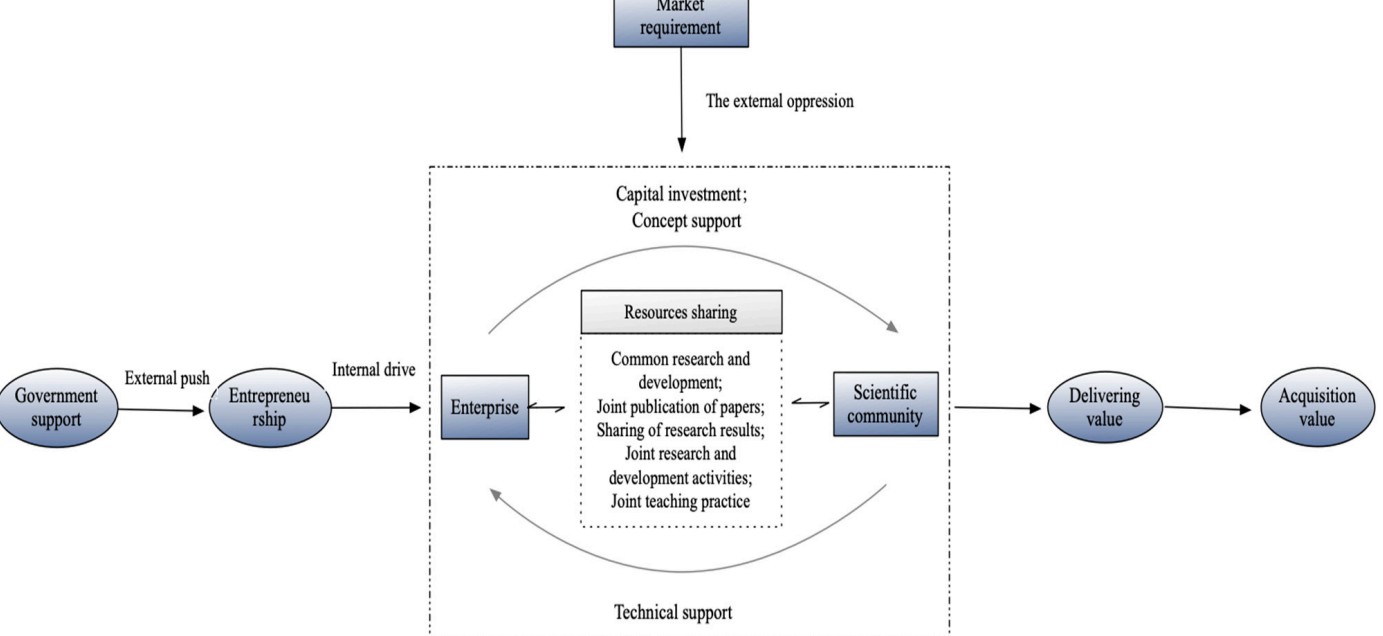

**Figure 3.** Enterprise–scientific community synergy value co-creation structure framework.

Value co-creation covers not only enterprise value and customer value but also the social, economic, political, cultural, and ecological value of all beneficiaries [48]. Value co-creation is an important factor for companies to build long-term customer relationships, achieve sustainable cash flow and drive innovative knowledge development [49]. In this case, Maoduoli shares the knowledge obtained from the cooperation with the scientific community through the museum and the special promotion of the supermarket. It propagates the brand culture, guides the green consensus, intuitively realizes the value transmission, and then gradually obtains the value (social (narrow sense), ecological, economic, and scientific research), such as tamarind resources protection, standardization planting, soil erosion observation governance, poverty alleviation, a national science and technology industry germplasm nursery construction, a comprehensive research angle of the whole development of the tamarind industrial chain, etc. In the end, the value of continuing feedback for enterprises and the scientific community is obtained and further promotes deeper cooperation development between enterprises and the scientific community; this is a process to create sustainable value and is shown in Figure 4.

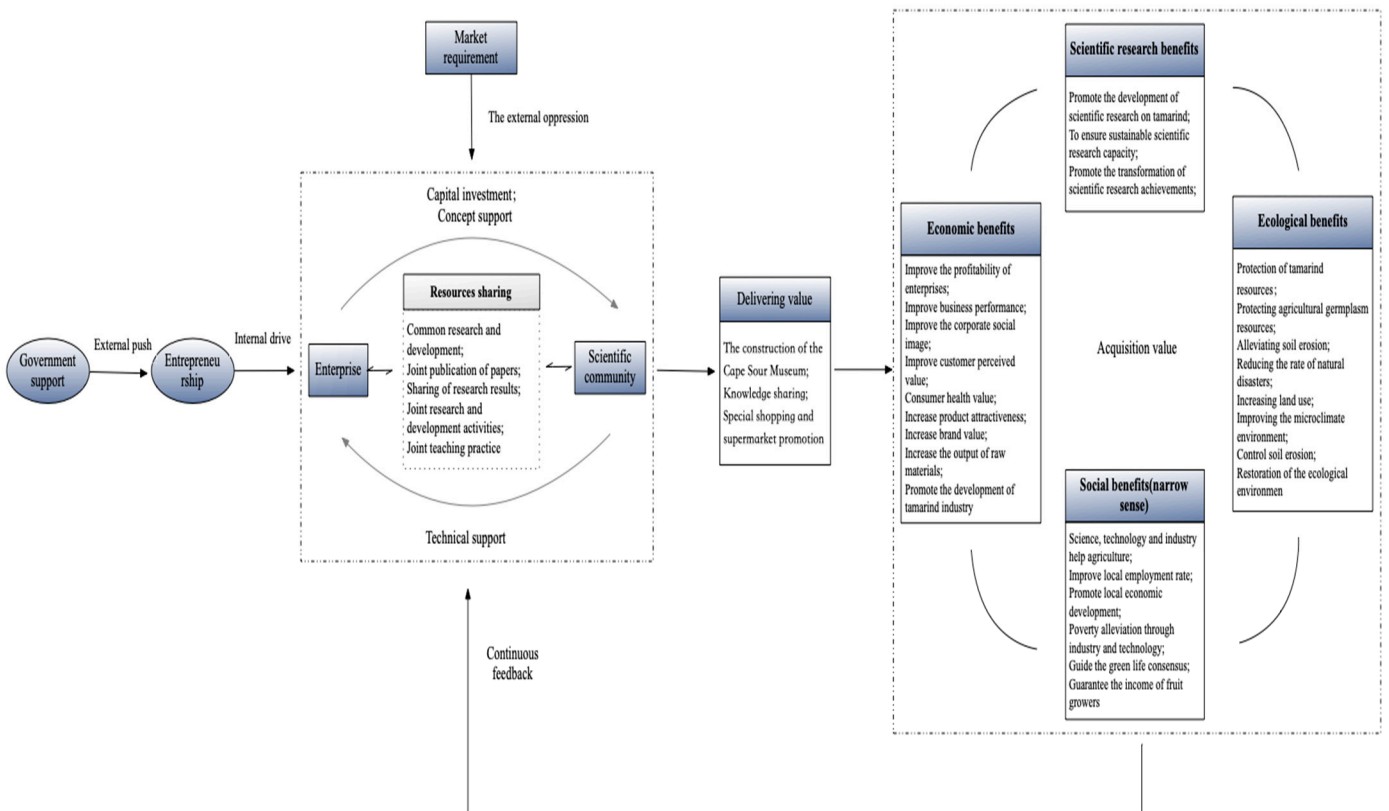

**Figure 4.** Model of cooperative value co-creation mechanism of enterprise–science community.

## 5. Epilogue

### 5.1. Theoretical Contribution

(1) The concept of the scientific community and related measurement items are introduced.

In the continuous in-depth analysis of the case, we found that under the background of deep integration of industry, university, and research, the scientific community is extremely prominent. Although existing studies have put forward the idea of value co-creation centered on stakeholders, there is no theoretical discussion on the value co-creation mode produced by the cooperative mechanism of enterprises and the scientific community. So, we introduced new constructs. This is the first time that the value co-creation mode under the collaborative mechanism has been explored by bundling the dimensions of enterprises and the scientific community, increasing the main dimension of the scientific community, and constructing its related measurement items.

(2) Construct and verify the structure model of the cooperative value co-creation mechanism between enterprises and the scientific community.

Although the existing pieces of literature interpret and analyze the value co-creation mechanism from different levels and angles, rarely, from a case perspective, do they explore the deep integration of industry, education, research, and value co-creation mode under the cooperative mechanism of enterprises and the scientific community. In this paper, through the in-depth analysis of Maoduoli and the scientific community and their collaborative value co-creation practice, the scientific community is placed in a parallel position with the two dimensions of consumers and stakeholders, forming special constructs, value-based logic, and the overall structural framework is constructed. It provides some theoretical support for enterprises to successfully practice a sustainable value co-creation mode under the cooperative mechanism of the scientific community and facilitates the exchange of knowledge and data between academia and practice.

*5.2. Practical Enlightenment*

Maoduoli's collaborative value co-creation model with the scientific community plays a crucial role in the rapid development of small and medium-sized enterprises from the small to micro stage. Its interactive activities with the scientific community, such as joint research and development, a joint publication of papers, sharing of research results, joint research and development activities, and joint teaching practice, are relevant to similar enterprises.

Maoduoli, in the process of delivering value, such as establishing the Cape Sour museum, provides a new idea for enterprise publicity. The construction of the Cape Sour museum can enhance the interaction between customers and also visually show the corporate culture, deepen the customer's understanding of the enterprise, and improve the customer's sense of identity. At the same time, the culture of tamarind can be popularized, and knowledge can be shared to benefit society. Therefore, the way publicity is represented by the display of culture and art may become a new starting point for enterprise publicity.

Maoduoli, with their scientific community collaborative value creating practice, is an excellent teaching case and management case sharing center for China, and is an outstanding practical case for Chinese enterprises to study the use of the national college teaching and encourage root university management practices. It is a line of research into enterprises, with case studies to refine management theory and teaching to cultivate new business talent. It facilitates the exchange of knowledge and data between academia and practice.

*5.3. Research Limitations and Future Prospects*

Although this paper has made a useful discussion on the collaborative value co-creation of the enterprise–science community, there are still some shortcomings that need further improvement in future research. First, in terms of research methods, this paper chose a single case study. Many case studies seem to have a weak foundation and some limitations. Therefore, we still need to obtain a more universally applicable specific theoretical proposition through the supplement and improvement of multiple case studies in the later stage. In addition, the analysis framework summarized by spindle coding analysis may have a suffocating effect on some original ideas in the coding process. Thirdly, although this paper extracts the mechanism model of collaborative value co-creation of the enterprise–science community from an overall perspective, there may be some new measurement items that are not included. Finally, under the national conditions with Chinese characteristics, the government and the public may also be important dimensions in carrying out the combined study.

Therefore, in future research, we will verify and supplement this model with multiple cases, select multiple enterprises to enrich this model, and further explore the main dimension of collaborative value co-creation under the national conditions of socialism with Chinese characteristics based on this model to improve this structural model. In addition, we find that the publicity represented by art and culture may become a new research direction for enterprise publicity.

**Author Contributions:** Conceptualization, W.S. and D.T.; methodology, W.S. and Q.Z.; software, Z.S. and Q.Z.; validation, Y.N. and Z.S.; formal analysis, W.S. and Y.N.; investigation, W.S. and C.L.; resources, W.S., C.L. and Y.N.; data curation, Z.S., Q.Z. and D.T.; writing—original draft preparation, W.S. and D.T.; writing—review and editing, W.S., Y.N., Z.S., C.L., Q.Z. and D.T.; visualization, Y.N., C.L., Q.Z. and D.T.; supervision, Y.N., Y.N., C.L. and D.T.; project administration, D.T. All authors have read and agreed to the published version of the manuscript.

**Funding:** This research received no external funding.

**Institutional Review Board Statement:** Not applicable.

**Informed Consent Statement:** Not applicable.

**Data Availability Statement:** Not applicable.

**Acknowledgments:** We thank anonymous commenters for their insightful comments on this manuscript, and the Maoduoli Group for timely help with research and interviews.

**Conflicts of Interest:** All the authors declare no conflicts of interest. Mr. Zibo Song was employed by Yunnan Mao Duo Li Group Food Co., Ltd., Kunming 650221, China. The remaining authors declare that the research was conducted in the absence of any commercial or financial relationships that could be construed as a potential conflict of interest.

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
