# Peer review of "Research on the Mechanism of Collaborative Value Co-Creation of Enterprise–Science Community: A Case Study Based on the Green Brand Maoduoli"

_sustainability, doi:10.3390/su142215439_

Round 1

Reviewer 1 Report

Thank you for the good research work. Please increase the practical implications of the study only. 

Author Response

Dear reviewer, thank you very much for your valuable comments and suggestions. According to your proposal, We have substantially revised the practical significance of the study so as to get your approval.

Point 1: Abstract: Please increase the practical implications of the study only.

Response 1: Thank you very much for your advice. We have increased the practical implications of the study to your suggestion. First of all, we supplemented the interactive activities of the enterprise science community for enterprises to learn from; Secondly, we have added the advantages and innovations of the new publicity method adopted by Maoduoli in the process of transferring value; Finally, we added that Maoduoli, as a teaching case, was included by the China Management Case Sharing Center, which is conducive to the training of new business talents in colleges and universities, and promotes the exchange of knowledge and data between academia and practice. Please see Lines 442-462.

Reviewer 2 Report

Dear authors, I read the paper with attention and interest. I found it very interesting and with novel aspects that improve the knowledge in the field. Indeed, it makes significant contribution to the body of knowledge related to this journal. Although the manuscript is interesting, it shows several weaknesses. Please pay attention to my comments below:

1. Abstract: Please clarify the objectives of the study and recommendation or suggestion of the research. The findings must be conclude in comprehensive writing. 

2. Citation: Please refer to the formatting/APA guideline for this journal. Please also check typo and many misspelling

3. Page 2-19 (65): The authors stress that the research on external innovation subjects is obviously small, and suddenly mention little research in external innovation entities, is it the same concept, or something that the authors try to relate with?

4. Page 2-19 (67): previous LR mention that external innovation subjects is very small, but the scope has gradually expanded. This statement is not clear and confused 

5. Page 2-19 (71): The authors mention about scientific community but did not mention clearly what is category/concept of this terms. 

6. Page 2-19 (83): the explanation on the synergy theory is not clear and the basic knowledge on this synergy is too scant 

7. Page 3-19 (106): The authors should cite the scholars or any previous researchers mentioned about the change to support the statement. 

8. Page 3-19 (138): before such as, use "," or can delete "such as'

9. Page 4-19 (147): consumer-centered to stakeholder-centered? the last centered for overall cycle or for value? 

10. Page 4-19 (183): the external innovation subject -- the scientific ... please check this

11. Page 5-19 (203): any references/citation or report that we can refer

12. Page 6-19 (232): justify why senior managers?

13. Page 7-19 (246): Paragraph 234 state the date from December 2020 to January 2021, but here January 2022? Please check the year to make sure in line with previous statement 

14. Page 8-19  (262): The authors should provided the process/cycle for open coding, spindle coding, selective coding - sorting and analysis, so that we know where (in figure) its happen 

15. Page 16-19 (458): attache -  typo?

15. Page 16-19 (462-466): Research limitations is too scant. Need more elaboration in terms of the process to finish, time etc. 

Author Response

Dear reviewer, thank you very much for your valuable comments and suggestions. According to your proposal, we have made a substantial revision of the paper from several aspects, such as abstract, introduction, relevant literature review, citation format, data source, grounded analysis process, research limitations so as to get your approval.

Point 1: Abstract: Please clarify the objectives of the study and recommendation or suggestion of the research. The findings must be conclude in comprehensive writing.

Response 1: Thank you very much for your advice. We have improved the summary according to your suggestion. The purpose and suggestions of this study are clarified. This study aims to explore how the enterprise-science community achieves sustainable collaborative value co-creation and explains how the enterprise-science community collaborative value co-creation mode is formed, how to implement it, how to deliver value to customers, and finally, the overall process of jointly harvesting value. Please see Lines 15-31.

Point 2: Citation: Please refer to the formatting/APA guideline for this journal. Please also check typo and many misspelling.

Response 2: Thank you very much for your advice. We have improved the citation format and the spelling as you suggested. We referred to the formatting/APA guideline for this journal and improved many misspellings.

Point 3: Page 2-19 (65): The authors stress that the research on external innovation subjects is obviously small, and suddenly mention little research in external innovation entities, is it the same concept, or something that the authors try to relate with?

Response 3: Thank you very much for your advice. We have improved the introduction as you suggested. We deleted language such as "less research on external innovation subjects", clarified the subject of "scientific community", and improved the literature review around the direction of enterprise-scientific community collaborative value co-creation. Please see Lines 53-68.

Point 4: Page 2-19 (67): previous LR mention that external innovation subjects is very small, but the scope has gradually expanded. This statement is not clear and confused

Response 4: Thank you very much for your advice. We have improved the introduction as you suggested. We deleted language, such as "external innovation subjects is very small, but the scope has gradually expanded", clarified the subject of "scientific community", and improved the literature review around the direction of enterprise-scientific community collaborative value co-creation. Please see Lines 53-68.

Point 5: Page 2-19 (71): The authors mention about scientific community but did not mention clearly what is category/concept of this terms.

Response 5: Thank you very much for your advice. We have improved the relevant literature review as you suggested. We have added the categories and composition of "scientific communities". Please see Lines 96-110.

Point 6: Page 2-19 (83): the explanation on the synergy theory is not clear and the basic knowledge on this synergy is too scant.

Response 6: Thank you very much for your advice. We have improved the relevant literature review as you suggested. We supplement the research on synergy theory and the theory of collaborative innovation of the enterprise-science community. Please see Lines 74-86.

Point 7: Page 3-19 (106): The authors should cite the scholars or any previous researchers mentioned about the change to support the statement.

Response 7: Thank you very much for your advice. We have improved the relevant literature review as you suggested. We summarized the changes in collaborative theory research from relevant literature and cited some representative relevant literature supplemented with evidence for the changes. Please see Lines 140-148.

Point 8: Page 3-19 (138): before such as, use "," or can delete "such as'

Response 8: Thank you very much for your advice. We have corrected the error statement, as you suggested. We have deleted the statement "such as".

Point 9:  Page 4-19 (147): consumer-centered to stakeholder-centered? the last centered for overall cycle or for value?

Response 9: Thank you very much for your advice. We have improved the relevant literature review as you suggested. We reintegrated the related literature on value co-creation of enterprise-science community, removed the phrase "change from consumer-centered to stakeholder-centered value co-creation mode", and briefly explained the two leading logics of value co-creation theory. Please see Lines 112-138.

Point 10: Page 4-19 (183): the external innovation subject -- the scientific ... please check this

Response 10: Thank you very much for your advice. We have improved the research methods as you suggested. We deleted this part of the statement related to the external innovation body and focused on the exploration of how can the enterprise-science community achieve sustainable collaborative value co-creation. Please see Lines 167-168.

Point 11: Page 5-19 (203): any references/citation or report that we can refer

Response 11: Thank you very much for your advice. We have supplemented the source information as you suggested. Please see Lines 187-189.

Point 12: Page 6-19 (232): justify why senior managers?

Response 12: Thank you very much for your advice. We have corrected the inaccurate statements as you suggested. We changed "senior manager" to "senior management", and our in-depth interview subjects were: the corporate chief financial officer, chairman of the board, base management, and so on. Therefore, they were collectively referred to as "senior management". We took photos during the in-depth interview for evidence. Please see Lines 212-213.

Point 13: Page 7-19 (246): Paragraph 234 state the date from December 2020 to January 2021, but here January 2022? Please check the year to make sure in line with previous statement

Response 13: Thank you very much for your advice. The period is correct. We did different surveys for each of the two time periods. In Line 216, from December 2020 to January 2021, we conduct a participatory observation of shopping and supermarket buyers. The period in Line 228 (December 2020 - January 2022) is the complete period of the case study process.

Point 14: Page 8-19 (262): The authors should provided the process/cycle for open coding, spindle coding, selective coding

Response 14: Thank you very much for your advice. We have improved the analysis process of spindle coding and selective coding according to your suggestion. Based on the open coding, spindle coding, and selective coding tables in the text, we further explained the refining and analysis process of the three tables in text form. Please see Lines 270-283 and Lines 289-296.

Point 15: Page 16-19 (458): attache - typo?

Response 15: Thank you very much for your advice. We have deleted the wrong sentence according to your suggestion.

Point 16: Page 16-19 (462-466): Research limitations is too scant. Need more elaboration in terms of the process to finish, time etc.

Response 16: Thank you very much for your advice. We have supplemented the research limitations according to your suggestions. We have added dimension defects and enriched future research directions. Please see Lines 464-483.

Reviewer 3 Report

The purpose of the paper is well-selected and relevant, the method is absolutely appropriate and it is well described. The problem statement is poor, there is not such a big gap in the literature about 'industry-university collaboration' (look up this keyword in Google Scholar and also check keywords industry-science collaboration, industry-science co-creation, industry-university co-creation) you will find a lot of studies worth reviewing. This will help you better formulate your research questions and the introductory problem statement. The introduction is a bit confused, sentences are long, try to be more up to the point and improve English.

The literature review section also needs to be improved and at present it is a bit confusing that you speak about synergy theory stakeholder centric view service-dominant logic etc. in short sub-sections. Try to merge some of them and be more up to the point: don't elaborate on everything but focus on industry-science collaboration and co-creation.

The definition of synergy is not correct, I recommend you don't use synergy theory but merely speak about resource sharing and leveraging the related synergy effect.

A major problem is with the Findings section. There are a lot of repetions here and more importantly, it is full of corporate slogans and non-scholarly adjectives. "the value co-creation of Maoduoli is extreme and enlightening" (?) 5/223 ; "Maoduoli's selfless dedication"? 14/389

Slogans and such praises need to be eradicated, instead, more should be written about the specifics of co-creation. For example you write somewhere that the company has 126 patents. Are these joint patents co-created by the collaboration partners (scientific institutes?)

Try to be more precise and describe facts and not sentences like that: Realizing that researchers are not getting enough 387 money for their research, Maoduoli feeds back on the research, expands and deepens the 388 research fields of researchers, and so on. 14/387-389

Author Response

Dear reviewer, thank you very much for your valuable comments and suggestions. According to your proposal, we have made a substantial revision of the paper from several aspects, such as introduction, literature review, case analysis and discovery, more precise academic statement so as to get your approval. 

Point 1: The introduction is a bit confused, sentences are long, try to be more up to the point and improve English

Response 1: Thank you very much for your advice. We have increased the practical implications of the study to your suggestion. We have improved the introduction according to your suggestion. We deleted redundant sentences, focused on the collaborative value creation of the enterprise science community, and reorganized the literature gap and research significance. Please see Lines 43-68.

Point 2: The literature review section also needs to be improved and at present it is a bit confusing that you speak about synergy theory stakeholder centric view service-dominant logic etc. in short sub-sections. Try to merge some of them and be more up to the point: don't elaborate on everything but focus on industry-science collaboration and co-creation.The definition of synergy is not correct, I recommend you don't use synergy theory but merely speak about resource sharing and leveraging the related synergy effect.

Response 2: Thank you very much for your advice. We have improved the literature review according to your suggestions. Focusing on the collaborative value and co-creation of the enterprise science community, we re-integrated the literature review, deleted redundant and invalid sentences, added industry-university cooperation and co-creation and other related literature content, supplemented the definition and structure of the science community and improved the literature review. Please see Lines 70-164.

Point 3: A major problem is with the Findings section. There are a lot of repetions here and more importantly, it is full of corporate slogans and non-scholarly adjectives. "the value co-creation of Maoduoli is extreme and enlightening" (?) 5/223 ;"Maoduoli's selfless dedication"? 14/389

Response 3: Thank you very much for your advice. We have improved the case analysis and discovery according to your suggestions. We have eliminated a large number of non-academic statements in this part. Please see Line 203 and Lines 316-382.

Point 4: Slogans and such praises need to be eradicated, instead, more should be written about the specifics of co-creation. For example you write somewhere that the company has 126 patents. Are these joint patents co-created by the collaboration partners (scientific institutes?). Try to be more precise and describe facts and not sentences like that: Realizing that researchers are not getting enough 387 money for their research, Maoduoli feeds back on the research, expands and deepens the 388 research fields of researchers, and so on. 14/387-389

Response 4: Thank you very much for your advice. We have improved the case analysis and discovery according to your suggestions. We have supplemented data, materials, and other relevant information to accurately describe the facts. Please see Lines 316-382.

Reviewer 4 Report

The authors of the article take up important issues of co-creating value by business and science. They say that this is a little known topic that requires research. And it is this kind of cooperation that can be treated as the foundation of the development of the economy.

It is a pity that the authors did not emphasize the fact that as part of science-business cooperation, added value can be created for both business and buyers.

The research is presented on the example of a case study. However, there is no justification for the selection of such and not another company. Although sustainable development is gaining more and more importance, it is not a sufficient criterion. It is imperative that the Authors justify their choice.

Besides, I have the impression that this is more a characteristic of the subject and the areas of its cooperation with science, rather than an actual indication of the processes of co-creating value for the client.

The article also does not provide information on how such cooperation should proceed in a model manner, and this would be particularly important from a practical point of view. Such a model could be an indication for other entities on how to start and conduct cooperation with business, so as to be able to co-create added value for buyers and stakeholders

Author Response

Dear reviewer, thank you very much for your valuable comments and suggestions. According to your proposal, we have made a substantial revision of the paper from several aspects, such as case selection, case analysis and discovery, practical enlightenment so as to get your approval. 

Point 1: It is a pity that the authors did not emphasize the fact that as part of science-business cooperation, added value can be created for both business and buyers.

Response 1: Thank you very much for your advice. We have improved the structural model according to your suggestions. We explained the process of "creating value - transferring value," how the case was conducted in enterprise science community cooperation, and how to create value for businesses and buyers. Please see Lines 245-247, Lines 268-281, Lines 289-296, and Lines 328-411.

Point 2: The research is presented on the example of a case study. However, there is no justification for the selection of such and not another company. Although sustainable development is gaining more and more importance, it is not a sufficient criterion. It is imperative that the Authors justify their choice.

Response 2: Thank you very much for your advice. We have improved the reasons for case selection according to your suggestions. With more precise language, we have rearranged the reasons for case selection, specifically in the following four aspects: 1. Compared with listed companies in the same food industry in China, the proportion of scientific research investment is extremely high; 2. Scientific research achievements (tamarind polysaccharide, etc.) are top in the world; 3. It has won many honors with high influence and is popular with consumers; 4. The practice of value creation in collaboration between Maoduoli and the scientific community, as an excellent teaching case, was included in the well-known academic platform - China Management Case Sharing Center (Case No.: MKT-1061) for reference by universities and enterprises across the country. It is a model for Chinese enterprises. Therefore, based on the principles of typicality and extremes, we have chosen such a representative enterprise. Please see Lines 177-206.

Point 3: Besides, I have the impression that this is more a characteristic of the subject and the areas of its cooperation with science, rather than an actual indication of the processes of co-creating value for the client.

Response 3: Thank you very much for your advice. We have supplemented the process and practical instructions of collaborative value co-creation of this model according to your suggestions. Following the value logic, we enriched the process of "creating value - transferring value - acquiring value", improved the model, and supplemented the practical significance of this study. Please see Lines 245-247, Lines 268-281, Lines 289-296, Lines 328-411, and Lines 442-462.

Point 4: The article also does not provide information on how such cooperation should proceed in a model manner, and this would be particularly important from a practical point of view. Such a model could be an indication for other entities on how to start and conduct cooperation with business, so as to be able to co-create added value for buyers and stakeholders

Response 4: Thank you very much for your advice. We have supplemented the specific contents of the enterprise science community collaborative value co-creation model according to your suggestions. We supplemented and explained how the enterprise science community collaborative value co-creation model was formed, how to implement it, how to deliver value to customers, and the overall process of finally harvesting value together. Please see Lines 245-247, Lines 268-281, Lines 289-296, and Lines 328-411.

Round 2

Reviewer 3 Report

The manuscript has been sufficiently improved. Thank you for addressing all my concerns.